# A Personalized 3D-Printed CAD/CAM Functional Space Maintainer Following the Premature Loss of a Primary First Molar in a Five-Year-Old Child

**DOI:** 10.3390/reports8030125

**Published:** 2025-07-29

**Authors:** Rasa Mladenovic, Andrija Nedeljkovic, Ljiljana Vujacic, Marko Stevanovic, Vladan Djordjevic, Srbislav Pajic, Kristina Mladenovic

**Affiliations:** 1Department for Dentistry, Faculty of Medical Sciences, University of Kragujevac, 34000 Kragujevac, Serbia; nedeljkovicandrija96@gmail.com (A.N.); drvladandjordjevic@gmail.com (V.D.); nevus-ng@hotmail.com (S.P.); 2Dental Medicine Clinic Dentokids, 34000 Kragujevac, Serbia; 3Department for Dentistry, Faculty of Medicine, University of Pristina, 38220 Kosovska Mitrovica, Serbia; titikakayu@yahoo.com (L.V.); marko.d.stevanovic@gmail.com (M.S.); 4Department of Physical Medicine and Rehabilitation, University Clinical Center of Kragujevac, 3400 Kragujevac, Serbia; kristinamladenovic1990@gmail.com; 5Department of Physical Medicine and Rehabilitation, Faculty of Medical Sciences, University of Kragujevac, 34000 Kragujevac, Serbia

**Keywords:** primary molar, tooth loss, space maintainers, CAD/CAM technology, 3D printing, masticatory performance

## Abstract

Primary teeth play a crucial role in a child’s development, particularly in maintaining space for permanent teeth. The premature loss of a primary tooth can lead to orthodontic issues, making the use of space maintainers essential to ensure proper growth and development of permanent teeth. To preserve space, the fabrication of a space maintainer is necessary. Since conventional space maintainers do not restore masticatory function, this study presents an innovative solution for space preservation following the extraction of the first primary molar through the design of the functional space maintainer KOS&MET (Key Orthodontic System and Materials Enhanced Therapy). The space maintainer was designed using the 3Shape Dental Designer 2023 version software tool and manufactured via additive 3D printing, utilizing a metal alloy with high resistance to masticatory forces. The crown is supported by the primary canine, while an intraoral window is created to monitor the eruption of the successor tooth. This design does not interfere with occlusion and enables bilateral chewing. Masticatory performance was assessed using two-color chewing gum, and the results showed improvement after cementing the space maintainer. This innovative approach not only preserves space for permanent teeth but also enhances masticatory function, contributing to the proper growth and development of the jaws and teeth.

Primary teeth play a vital role in a child’s growth and development—not only in functions such as chewing, speech, and esthetics, but also in the proper formation of the jaws, maintenance of occlusion, and preservation of space for the eruption of permanent teeth [1]. The premature loss of even a single primary tooth can lead to disturbances in the dental arch, shifting of adjacent teeth, malocclusion, and prolonged orthodontic treatments in the future. Additionally, impaired chewing caused by tooth loss can negatively affect the development of facial and jaw muscles, the function of the temporomandibular joint, and the child’s nutrition and overall health [2,3]. To prevent these complications, the use of space maintainers plays an essential role, as they enable the uninterrupted eruption of permanent teeth and the preservation of proper dental arch structure [4,5]. In any case, space maintainers must meet several key criteria: they must maintain the mesiodistal width of the lost tooth and be cost-effective, simple to fabricate and apply, strong enough to withstand masticatory forces, and easy to maintain. Advancements in digital dentistry, including CAD/CAM technology and 3D printing [6], have further improved the precision and customization of space maintainers, allowing for better adaptation to the individual anatomy of each child.

A five-year-old boy was referred to the Department of Dentistry at the Faculty of Medical Sciences in Kragujevac. He was generally healthy, with no significant medical history. The first primary molar was extracted due to complications of caries, and his oral hygiene was satisfactory. No pre-existing occlusal issues were noted. Due to the premature tooth loss, an innovative functional space maintainer was planned (Figure 1, Figure 2 and Figure 3).

This result indicates an improvement in chewing function and the effectiveness of the space maintainer after cementing. This innovative approach to the creation of space maintainers contributes not only to preserving space for permanent teeth but also to improving chewing function, providing long-term benefits in the development of the child’s dental health. Additionally, this approach allows for precise evaluation and monitoring of the development of permanent teeth, ensuring optimal therapy for the preservation of oral health and function throughout the child’s development.

A particular advantage of CAD/CAM space maintainers lies in the digital nature of the fabrication process. Unlike conventional Band and Loop space maintainers [1,6], which require manual impression taking and laboratory work, CAD/CAM space maintainers are designed and produced using precise digital workflows. This allows for greater customization, improved fit, and faster fabrication times. Additionally, in case of damage or the need for replacement, the digital model can be reused without requiring new impressions, reducing discomfort for the child and overall treatment time. These features enhance clinical efficiency and patient comfort, especially important in pediatric dentistry. As a limitation of this case report, we note the 12-month follow-up period; therefore, it is necessary to apply this type of space maintainer in more clinical cases with extended follow-up durations. Additionally, further studies are needed to confirm the applicability of this design in cases of second primary molar loss, when the first permanent molar is already present in the oral cavity.

## Figures and Tables

**Figure 1 reports-08-00125-f001:**
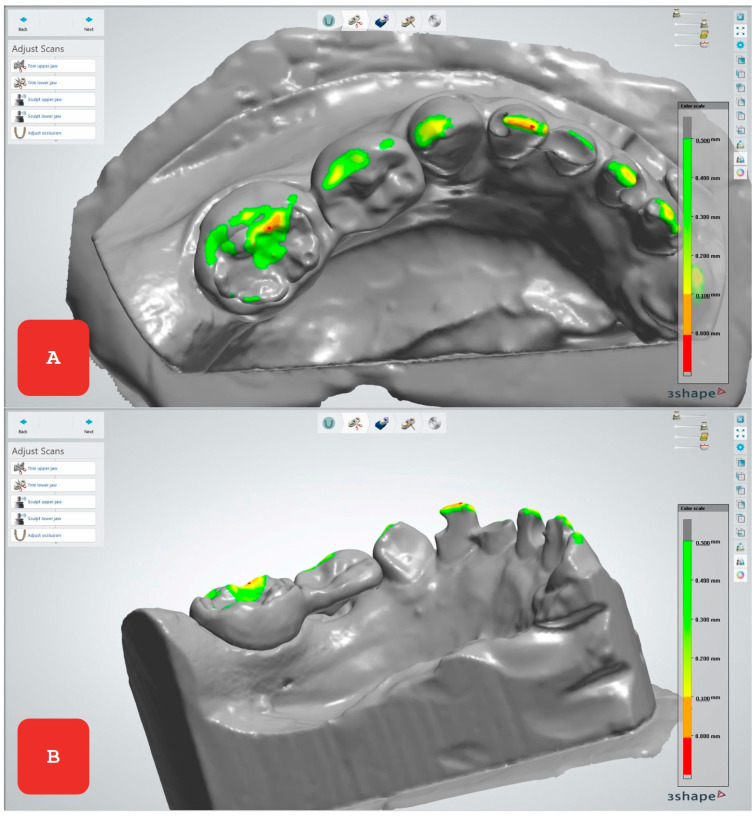
We used 3Shape Dental Designer (3Shape A/S, Copenhagen, Denmark) to model the space maintainer precisely according to the patient’s needs. Since the first primary molar was lost, we designed the space maintainer based on a band and suspended crown by Mladenovic, naming it KOS&MET (Key Orthodontic System and Materials Enhanced Therapy). The goal of this design was not only to preserve space but also to restore bilateral chewing function in the child. The crown replacing the missing first molar rests on the distal surface of the primary canine, providing stability and functionality, while an intraoral window was created to allow clinical assessment of the eruption of the successor tooth and timely removal of the space maintainer. For the morphology of the missing primary molar, we used our own library of previously scanned primary molars, which allowed precise modeling of the crown that perfectly fits the natural position in the child’s dental arch (**A**). The band surrounding the second primary molar was designed not to interfere with occlusion or cause bite issues (**B**). This design allows for normal occlusal function and efficient chewing, which is crucial for the development of the child’s oral function during growth.

**Figure 2 reports-08-00125-f002:**
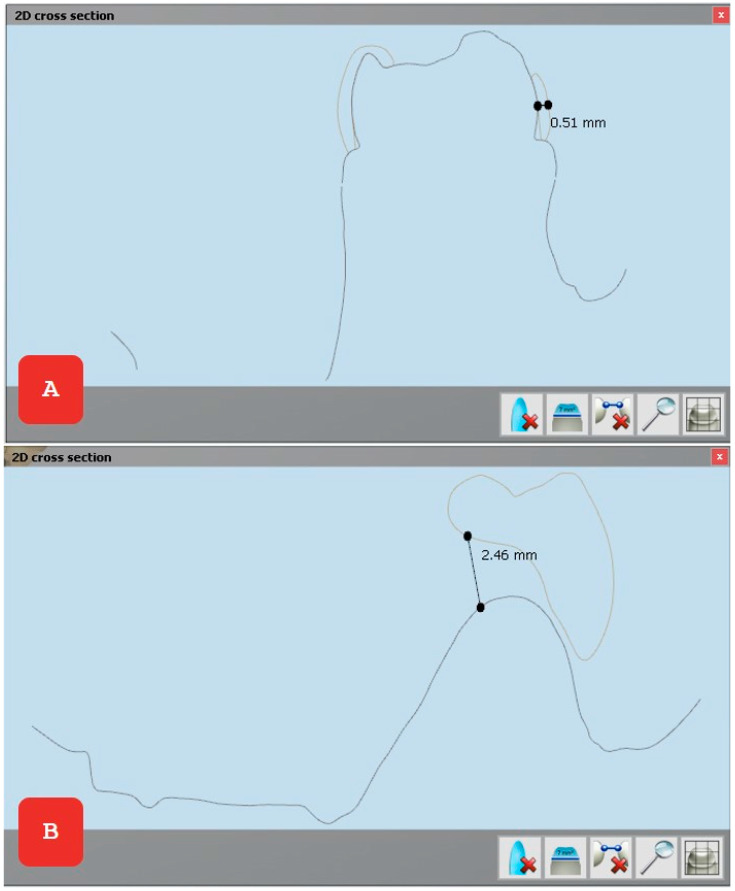
The band of the KOS&MET functional space maintainer is less than 1 mm in width (**A**), allowing optimal adaptation within the dental arch without interfering with the normal growth and development of surrounding structures. The intraoral window is over 2 mm wide (**B**), enabling easy monitoring of the eruption of the replacement tooth and facilitating oral hygiene maintenance, thereby preventing potential infections and complications during treatment.

**Figure 3 reports-08-00125-f003:**
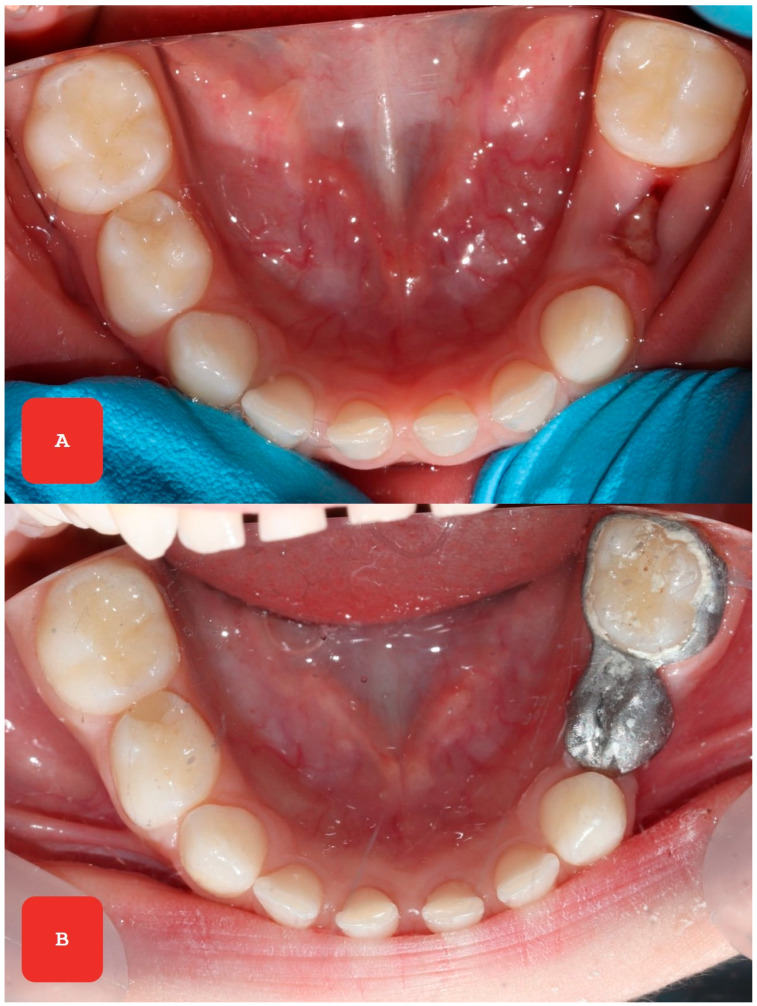
Following the premature extraction of the first primary molar (**A**), the KOS&MET functional space maintainer was fabricated from a metal alloy (**B**) (Co 69%, Cr 25%, W 9.5%, Mo 3.5%, Si 1%, Scheftner, Germany) using the additive 3D printing method (MySint100 Dual Laser, Sisma S.p.A., Piovene Rocchette, Italy). This technology allows for precise fabrication and high resistance to masticatory forces, which is important for the long-term preservation of function. Glass ionomer cement (3M ESPE Ketac Cem, Saint Paul, Minnesota, USA) was used for cementing the space maintainer, providing adequate stability and fixation over time. After cementing, the space maintainer was monitored through regular check-ups at 1, 3, 6, and 12 months, with no detachment or breakage observed. Additionally, there were no changes in the soft tissues or the supporting tooth, which further confirmed the success of this method. To assess masticatory performance, a two-color chewing gum (Hue-Check Gum©, University of Bern, Bern, Switzerland) was used. The result was evaluated using the Orophys scale, which categorizes mixing into five levels [7]. Before chewing, the values were SA2 (large portions of chewing gum were not mixed), while after cementing, the value was SA4 (the bolus was well mixed, but the color was not uniform).

## Data Availability

The original contributions presented in this study are included in the article. Further inquiries can be directed to the corresponding author.

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
