# Peer review of "A Personalized 3D-Printed CAD/CAM Functional Space Maintainer Following the Premature Loss of a Primary First Molar in a Five-Year-Old Child"

_reports, 2025, doi:10.3390/reports8030125_

Round 1

Reviewer 1 Report

Comments and Suggestions for Authors

Dear authors,

I appreciate the authors' efforts in conducting this research and submitting their work for consideration. The report is well-structured and clearly written. However, some points need clarification before considering publication.

  1. The abstract states “This study presents an innovative solution...”, but the IRB Statement clarifies that this is “one clinically treated patient whose dental treatment was not originally intended to be the subject of a report”. Also, the title already indicates “a Five-Year-Old Child”.
  2. Modify the abstract and introduction to state that the paper presents a case report rather than a study
  3. The authors mentioned 12 months of follow-up which is good for initial assessment; However, the paper frequently mentions long-term benefits. longer-term follow-up data (beyond 12 months and until permanent tooth eruption) would be beneficial to demonstrate sustained efficacy.
  4. Add a limitations section to discuss the limitations in a single-case report, particularly regarding generalizability and the current duration of follow-up.
  5. The Orophys scale, while useful, categorizes mixing into only 5 levels. While improvement was noted (SA2 to SA4), however, if the authors will consider incorporating quantitative/qualitative measures would be more objective for masticatory efficiency.
  6. The authors list cost-effectiveness as a key criterion for space maintainers. While CAD/CAM and 3D printing offer precision and personalization, their cost implications compared to conventional methods are not discussed. Please add a brief discussion on this aspect.
  7. It would be valuable if the authors provided more clinical details about the child (specific reason for extraction beyond complications of caries, initial oral hygiene status, any pre-existing occlusal issues to enhance the case presentation without compromising confidentiality.
  8. Figures 1 and 2 are presented with descriptive text but without direct or specific discussion points linking them into the flow of the main text. Ensure each figure is referenced and discussed where relevant in the narrative.

Author Response

Dear authors,

I appreciate the authors' efforts in conducting this research and submitting their work for consideration. The report is well-structured and clearly written. However, some points need clarification before considering publication.

  1. The abstract states “This study presents an innovative solution...”, but the IRB Statement clarifies that this is “one clinically treated patient whose dental treatment was not originally intended to be the subject of a report”. Also, the title already indicates “a Five-Year-Old Child”.
  • ANSWER: Corrected

  1. Modify the abstract and introduction to state that the paper presents a case report rather than a study
  • ANSWER: Corrected

  1. The authors mentioned 12 months of follow-up which is good for initial assessment; However, the paper frequently mentions long-term benefits. longer-term follow-up data (beyond 12 months and until permanent tooth eruption) would be beneficial to demonstrate sustained efficacy.
  • ANSWER: We believe that a 12-month period is sufficient for assessing clinical sustainability. Since the space maintainer is still active in the child’s mouth, we are currently unable to evaluate it over a longer period.

  1. Add a limitations section to discuss the limitations in a single-case report, particularly regarding generalizability and the current duration of follow-up.
  • ANSWER: Corrected

  1. The Orophys scale, while useful, categorizes mixing into only 5 levels. While improvement was noted (SA2 to SA4), however, if the authors will consider incorporating quantitative/qualitative measures would be more objective for masticatory efficiency.
  • ANSWER: We chose the Orophys SA scale because it is easy to use and appealing to children, as it involves chewing gum.

  1. The authors list cost-effectiveness as a key criterion for space maintainers. While CAD/CAM and 3D printing offer precision and personalization, their cost implications compared to conventional methods are not discussed. Please add a brief discussion on this aspect.
  • ANSWER: Added

  1. It would be valuable if the authors provided more clinical details about the child (specific reason for extraction beyond complications of caries, initial oral hygiene status, any pre-existing occlusal issues to enhance the case presentation without compromising confidentiality.
  • ANSWER: Corrected

  1. Figures 1 and 2 are presented with descriptive text but without direct or specific discussion points linking them into the flow of the main text. Ensure each figure is referenced and discussed where relevant in the narrative.
  • ANSWER: Corrected

Reviewer 2 Report

Comments and Suggestions for Authors

The manuscript titled "Personalized 3D-Printed CAD/CAM Functional Space Maintainer Following Premature Loss of a Primary First Molar in a Five-Year-Old Child" introduces a new orthodontic space maintenance appliance. While the approach is novel it is difficult to evaluate due to organization of the manuscript and missing information. A clear design needs to be articulated, for an example is this a case study versus a study?

I do believe that a case report is a more appropriate presentation format for this submission and would provide more information and context for the reader.

In addition, the manuscript is in need of structure and additional information in the following order:

  • Introduction/background/purpose
  • Case Presentation (i.e., fabrication of appliance, materials, patient information, patient outcomes, challenges, etc.)
  • Discussion (comparison to existing care models, advantages vs disadvantages, cost, etc.)
  • Conclusion
  • The abstract should be presented in a similar manner. Figures should be appropriately labeled and reference in the manuscript.

Author Response

The manuscript titled "Personalized 3D-Printed CAD/CAM Functional Space Maintainer Following Premature Loss of a Primary First Molar in a Five-Year-Old Child" introduces a new orthodontic space maintenance appliance. While the approach is novel it is difficult to evaluate due to organization of the manuscript and missing information. A clear design needs to be articulated, for an example is this a case study versus a study?

I do believe that a case report is a more appropriate presentation format for this submission and would provide more information and context for the reader.

In addition, the manuscript is in need of structure and additional information in the following order:

  • Introduction/background/purpose
  • Case Presentation (i.e., fabrication of appliance, materials, patient information, patient outcomes, challenges, etc.)
  • Discussion (comparison to existing care models, advantages vs disadvantages, cost, etc.)
  • Conclusion
  • The abstract should be presented in a similar manner. Figures should be appropriately labeled and reference in the manuscript.
  • ANSWER: In accordance with the guidelines of the journal Reports for writing an Interesting Image, we are unable to revise the entire manuscript. Necessary changes have been made to the organization of the article. A complete revision of the entire manuscript, including your comments, has also been carried out.

Reviewer 3 Report

Comments and Suggestions for Authors

The manuscript is well structured, the images are informative, and the outcomes are convincingly presented.

To further align the submission with the scope of Reports, I recommend a brief addition discussing the rationale for using a functional space maintainer in preschool-aged patients, including its potential developmental advantages compared to conventional passive space maintainers. A short paragraph contrasting this device with traditional options (e.g., band-and-loop) in terms of clinical indications, functional benefits, and adaptability in similar pediatric cases would strengthen the broader relevance of the report.

Author Response

The manuscript is well structured, the images are informative, and the outcomes are convincingly presented.

To further align the submission with the scope of Reports, I recommend a brief addition discussing the rationale for using a functional space maintainer in preschool-aged patients, including its potential developmental advantages compared to conventional passive space maintainers. A short paragraph contrasting this device with traditional options (e.g., band-and-loop) in terms of clinical indications, functional benefits, and adaptability in similar pediatric cases would strengthen the broader relevance of the report.

  • ANSWER: Thank you for the comments. We have added a section about Band and Loop.

Reviewer 4 Report

Comments and Suggestions for Authors

Thank you for the opportunity to review the manuscript. Although this is classified as an "Interesting Image," I highly recommend rewriting it as a single text rather than separate descriptions of individual figures.

My suggestions:

  1. Though such a type of work can go without separate sections such as introduction, methodology, results, discussion, and conclusions, the manuscript should have a short introductory paragraph presenting the problem and justifying the novelty and importance of the clinical case described.
  2. When presenting the case, references to the latest literature should be provided. Try to include references that are no more than 10 years old, and please expand your reference list.
  3. Figures should be presented sequentially in the text itself. If the figure consists of several parts, I recommend marking each of them with the letters "a, b, c" and writing what is shown in each of these parts in the figure caption.
  4. Please, compare your proposed device with other non-removable space maintainers described in the literature and highlight the advantages of your device. Also, discuss whether it can be manufactured after the loss of the second primary molar.
  5. When presenting the construction of your device, indicate whether it touches the distal contact edge of the primary canine.
  6. For keywords, it is recommended to use words/phrases that are not directly in the title. This increases the likelihood of finding the publication during the search.

I encourage resubmission of the manuscript once all comments have been addressed.

Author Response

Thank you for the opportunity to review the manuscript. Although this is classified as an "Interesting Image," I highly recommend rewriting it as a single text rather than separate descriptions of individual figures.

My suggestions:

  1. Though such a type of work can go without separate sections such as introduction, methodology, results, discussion, and conclusions, the manuscript should have a short introductory paragraph presenting the problem and justifying the novelty and importance of the clinical case described.
  • ANSWER: Corrected
  1. When presenting the case, references to the latest literature should be provided. Try to include references that are no more than 10 years old, and please expand your reference list.
  • ANSWER: Corrected
  1. Figures should be presented sequentially in the text itself. If the figure consists of several parts, I recommend marking each of them with the letters "a, b, c" and writing what is shown in each of these parts in the figure caption.
  • ANSWER: Corrected
  1. Please, compare your proposed device with other non-removable space maintainers described in the literature and highlight the advantages of your device. Also, discuss whether it can be manufactured after the loss of the second primary molar.
  • ANSWER: Corrected
  1. When presenting the construction of your device, indicate whether it touches the distal contact edge of the primary canine.
  • ANSWER: Corrected
  1. For keywords, it is recommended to use words/phrases that are not directly in the title. This increases the likelihood of finding the publication during the search.
  • ANSWER: Corrected

I encourage resubmission of the manuscript once all comments have been addressed.

Round 2

Reviewer 2 Report

Comments and Suggestions for Authors

The manuscript is improved, and concerns have been addressed.

Author Response

Thank you.

Reviewer 4 Report

Comments and Suggestions for Authors

Thank you for the opportunity to review the revised manuscript. The authors only partially took the comments into account and revised the manuscript accordingly.

I repeat my comments, which were not taken into account but which I believe are important:

  1. Figures should be presented consecutively in the text with references to them. If a figure consists of several parts, I recommend marking each of them with the letters "a, b, c" and writing in the figure caption what is depicted in each of these parts.
  2. Please discuss whether your described device can be manufactured after the loss of the second primary molar. This would reveal the versatility of the device.
  3. When presenting the construction of your device, indicate whether it touches the distal contact edge of the primary canine.
  4. For keywords, it is recommended to use words/phrases that are not directly in the title. This increases the likelihood of finding the publication during the search.
  5. Make sure the font size is consistent throughout. Your manuscript should read like a single piece of text, not a collection of separate paragraphs.

Please read the questions carefully and only resubmit your manuscript once all questions have been answered.

Author Response

REVIEWER 4

Thank you for the opportunity to review the revised manuscript. The authors only partially took the comments into account and revised the manuscript accordingly.

- ANSWER: Dear reviewer, in accordance with all your comments, we have revised the manuscript. Your feedback was very helpful in making the paper easier to read. Thank you once again.

I repeat my comments, which were not taken into account but which I believe are important:

  1. Figures should be presented consecutively in the text with references to them. If a figure consists of several parts, I recommend marking each of them with the letters "a, b, c" and writing in the figure caption what is depicted in each of these parts.

- ANSWER: We have revised the figures and integrated them within the descriptions and the main text.

  1. Please discuss whether your described device can be manufactured after the loss of the second primary molar. This would reveal the versatility of the device.

- ANSWER: We have added a section discussing the potential application in cases of second primary molar loss.

  1. When presenting the construction of your device, indicate whether it touches the distal contact edge of the primary canine.

- ANSWER: The information that it rests on the primary molar was already included, but we have now further revised this part of the text.

  1. For keywords, it is recommended to use words/phrases that are not directly in the title. This increases the likelihood of finding the publication during the search.

- ANSWER: We have added new keywords.

  1. Make sure the font size is consistent throughout. Your manuscript should read like a single piece of text, not a collection of separate paragraphs.

- ANSWER: We have revised the entire manuscript.

Please read the questions carefully and only resubmit your manuscript once all questions have been answered.

Round 3

Reviewer 4 Report

Comments and Suggestions for Authors

Thank you for your corrections.